# Prudent-Active and Fast-Food-Sedentary Dietary-Lifestyle Patterns: The Association with Adiposity, Nutrition Knowledge and Sociodemographic Factors in Polish Teenagers—The ABC of Healthy Eating Project

**DOI:** 10.3390/nu10121988

**Published:** 2018-12-15

**Authors:** Lidia Wadolowska, Jadwiga Hamulka, Joanna Kowalkowska, Malgorzata Kostecka, Katarzyna Wadolowska, Renata Biezanowska-Kopec, Ewa Czarniecka-Skubina, Witold Kozirok, Anna Piotrowska

**Affiliations:** 1Department of Human Nutrition, Faculty of Food Science, University of Warmia and Mazury in Olsztyn, Pl. Cieszynski 1, 10-718 Olsztyn, Poland; lidia.wadolowska@uwm.edu.pl (L.W.); joanna.kowalkowska@uwm.edu.pl (J.K.); kwadolow@gmail.com (K.W.); 2Department of Human Nutrition, Faculty of Human Nutrition and Consumer Sciences, Warsaw University of Life Science (SGGW-WULS), 159C Nowoursynowska Street, 02-787 Warsaw, Poland; jadwiga_hamulka@sggw.pl; 3Department of Chemistry, Faculty of Food Science and Biotechnology, University of Life Sciences, 15 Akademicka Street, 20-950 Lublin, Poland; 4Department of Human Nutrition, Faculty of Food Technology, University of Agriculture in Krakow, 122 Balicka Street, 30-149 Krakow, Poland; r.biezanowska-kopec@ur.krakow.pl; 5Department of Food Gastronomy and Food Hygiene, Faculty of Human Nutrition and Consumer Sciences, Warsaw University of Life Science (SGGW-WULS), 159C Nowoursynowska Street, 02-787 Warsaw, Poland; ewa_czarniecka_skubina@sggw.pl; 6Department of Commodity and Quality Management, Faculty of Entrepreneurship and Quality Science, Gdynia Maritime University, 81-87 Morska Street, 81-225 Gdynia, Poland; w.kozirok@wpit.umg.edu.pl; 7Department of Functional Food, Ecological Food and Commodities, Faculty of Human Nutrition and Consumer Sciences, Warsaw University of Life Science (SGGW-WULS), 159C Nowoursynowska Street, 02-787 Warsaw, Poland; anna_piotrowska@sggw.pl

**Keywords:** adolescent, cluster analysis, dietary patterns, food frequency questionnaire, nutrition knowledge, physical activity, screen time, sedentary behaviour, socioeconomic factors

## Abstract

A holistic approach to understanding the relationship between diet, lifestyle and obesity is a better approach than studying single factors. This study presents the clustering of dietary and lifestyle behaviours to determine the association of these dietary-lifestyle patterns (DLPs) with adiposity, nutrition knowledge, gender and sociodemographic factors in teenagers. The research was designed as a cross-sectional study with convenience sampling. The sample consisted of 1549 Polish students aged 11–13 years. DLPs were identified with cluster analysis. Logistic regression modelling with adjustment for confounders was applied. Three dietary-lifestyle patterns were identified: Prudent-Active (29.3% of the sample), Fast-food-Sedentary (13.8%) and notPrudent-notFast-food-lowActive (56.9%). Adherence to Prudent-Active pattern (reference: notPrudent-notFast-food-lowActive) was 29% or 49% lower in 12-year-old or 13-year-old teenagers than in 11-year-old teenagers, respectively, and higher by 57% or 2.4 times in the middle or the upper tertile than the bottom tertile of the nutrition knowledge score. To the contrary, adherence to Fast-food-Sedentary (reference: notPrudent-notFast-food-lowActive) was lower by 41% or 58% in the middle or the upper tertile than the bottom tertile of the nutrition knowledge score, respectively. In Prudent-Active, the chance of central obesity (waist-to-height ratio ≥0.5) was lower by 47% and overweight/obesity was lower by 38% or 33% (depending on which standard was used: International Obesity Task Force, 2012: BMI (body mass index)-for-age ≥ 25 kg/m^2^ or Polish standards, 2010: BMI-for-age ≥ 85th percentile) when compared with the notPrudent-notFast-food-lowActive pattern. In Fast-food-Sedentary, the chance of central obesity was 2.22 times higher than the Prudent-Active pattern. The study identified a set of characteristics that decreased the risk of general and central adiposity in teenagers, which includes health-promoting behaviours related to food, meal consumption and lifestyle. Avoiding high-energy dense foods is insufficient to prevent obesity, if physical activity and the consumption frequency of health-promoting foods are low and breakfast and a school meal are frequently skipped. The results highlight the importance of the nutrition knowledge of teenagers in shaping their health-promoting dietary habits and active lifestyle to decrease adiposity risk and negative aspects of lower family affluence which promotes unhealthy behaviours, both related to diet and lifestyle.

## 1. Introduction

Adolescence is a period of important changes in physical, mental and social development. The awareness of one’s own sexuality and gender differences arises with regard to nutritional and caloric needs as well as physical abilities. The precise indication of the age at which changes start in the perception of oneself as an independent individual and food consumer is difficult, but the age of 10–12 years old is considered a breakthrough [1]. In general, girls tend to be more knowledgeable about food, nutrition and health and they exhibit more positive health- and food-related beliefs than boys [2,3,4]. On the other hand, boys spend more time being active, while girls spend more time socializing [5,6,7]. In this period, the establishment of health-promoting dietary and lifestyle habits plays a crucial role for achieving full growth potential and preventing diet-related diseases throughout the whole lifespan [8,9].

Progressive globalization, rapid changes in the ways of spending leisure time and increasing access to food create an adiposity-promoting environment with two main factors affecting the positive energy balance and obesity—sedentary lifestyle and an unhealthy diet rich in food whose consumption should be limited [9]. In real life, there is no single factor responsible for obesity. Moreover, adiposity-promoting behaviours can exist together with health-promoting behaviours because young people can engage in a variety of behaviours. It was reported by Gubbels et al. [10] that some boys combined unhealthy with health-promoting dietary and lifestyle behaviours. The pattern labelled as ‘Sports-Computer’ was characterized by a lower frequency of take-out meals and higher frequency of eating meals together, spending more time sitting and using a computer and also being physically active. A comprehensive approach, combining many environmental exposures is a better way to study the causes of obesity rather than to consider single behaviours influencing health [11]. Furthermore, by identifying factors associated with behavioural patterns, high-risk groups can be found for preventive interventions to stop the growth of obesity-related non-communicable diseases [12,13].

There have been several studies focused on the clustering of dietary-related and lifestyle-related behaviours in teenagers [2,10,13,14,15,16,17,18,19,20,21,22,23,24]. Two main patterns, one described as an ‘active and healthy lifestyle’ and the opposite as a ‘high screen time and unhealthy lifestyle’, were reported in school-age children from Spain [14], Finland [15] and nine European countries [25]. Some differences were reported between these patterns, depending on the region and the most characteristic dietary habit for the population under study. To date, few studies have considered selected dietary and/or lifestyle variables in relation to overweight or obesity [14,15,16,17,19,20,25]. The variables mainly referred to time spent in front of the screen, sleep duration, sleep quality, smoking, chosen physical activities, meal consumption or accompanying circumstances [10,14,15,17]. For dietary characteristics, only the consumption of fruit, vegetables and fruit juices was considered [17,18]. Thus, the dietary component of dietary-lifestyle patterns is weakly known in teenagers, especially in relation to obesity.

The aim of the study was the clustering of dietary and lifestyle behaviours to determine the association of these dietary-lifestyle patterns (DLPs) with adiposity, nutrition knowledge, gender and sociodemographic factors in teenagers.

## 2. Methods

### 2.1. Study Design and Participants

The analyses related to this paper were designed as a cross-sectional study with convenience sampling. Analyses were carried out on data from the ABC of Healthy Eating national multicentre project and the research was carried out in parallel with the project by academic researchers. The data was collected (2015–2016) by well-trained researchers from nine Polish universities, in ten locations covering the entire territory of Poland. More details on the study protocol, sample collection, sample size calculation and methods have been described previously [26].

Recruitment was conducted in elementary schools from urban, sub-urban and rural areas. Students from fourth and fifth grade classes were invited to attend. In total, 116 classes were selected from across Poland (Figure 1). Initially, 1678 students were recruited. One hundred and twenty-nine participants were excluded from analyses: 109 participants because of an age below 11 years or above 13 years and 20 participants due to missing data on at least one dietary or lifestyle component. Finally, the study included 1549 teenagers 11–13 years old (48.5% boys).

### 2.2. Data Collection

Data related to diet, lifestyle, nutrition knowledge and socio-demographic factors were collected using a 50-item, short form of the food frequency questionnaire (SF-FFQ4PolishChildren) developed by Kowalkowska, Wadolowska and Hamulka for the ‘ABC of Healthy Eating’ project. The reproducibility of the short form of the FFQ was carried out in cooperation with researchers from 12 universities covering the entire territory of Poland (data not published, paper in preparation). In brief, the internal compatibility of the questionnaire was tested by 630 teenagers (11–15 years old). The questionnaire was completed by teenagers twice (test and retest after 2 weeks). For the measures under study, Fleiss’ Kappa was 0.30 to 0.89 and compatible classification of subjects (into the same category in test and retest) was 42.9% to 95.8%. Thus, the internal compatibility of the short form of the FFQ dedicated for teenagers was considered acceptable to very good.

The questionnaire was self-administered by teenagers in the classroom and supervised by well-trained researchers. Explanations were given if necessary. More details on this questionnaire were previously provided [26].

### 2.3. Dietary Data

Respondents specified their usual frequency of consumption within the 12 last months of: breakfast, a school meal and nine food items: dairy products, fish, vegetables, fruit, fruit or vegetable juices, fast foods, sweetened drinks, energy drinks and sweets. Breakfast and a school meal were both defined as consuming solid foods with or without beverages: breakfast—the first eating episode per day; a school meal—the second eating episode of the day while at school [27]. Drinking only beverages was not considered as breakfast or a school meal. More details on dietary data were previously provided [26]. In brief, respondents could choose one of the following:four categories of breakfast consumption (number of days/week): 0/week, 1–3/week, 4–6/week, 7/week (every day);four categories of school meal consumption (number of days/week): 0/week, 1–3/week, 3–4/week, 5/week (every school day);seven categories for food frequency consumption: never or almost never, less than once a week, once a week, 2–4 times/week, 5–6 times/week, every day, several times a day; for further analysis, all frequencies of food consumption were converted into daily frequency (details in Appendix A).

### 2.4. Lifestyle

Three measures of lifestyle were established—screen time, physical activity at school and physical activity at leisure time.

Screen time was assessed using the question ‘How much time do you spend watching TV or on the computer or in front of a computer on an average day of the week?’ The respondents could choose one of six answers: <2 h/day, 2 to <4 h/day, 4 to <6 h/day, 6 to <8 h/day, 8 to <10 h/day, ≥10 h/day. To express screen time in points, answers were converted into scores (1 to 6 points) (Appendix A).

To describe physical activity at school or leisure time, the participants could choose one of three answers (low, moderate, vigorous). Many examples for each answer were given (Appendix A). To express both physical activities in points, answers were converted into scores (both 1 to 3 points).

### 2.5. Dietary-Lifestyle Patterns

Dietary-lifestyle patterns were derived using cluster analysis. The input variables were 11 dietary (in times/day) and 3 lifestyle (in scores) components of DLPs. All input variables were standardized, using our own database, to achieve a mean equal to 0 and a standard deviation equal to 1. To identify the optimal number of clusters, the analysis was conducted several times. The K-means clustering algorithm was used and subjects were grouped based on the Euclidean distances. Finally, three clusters were selected (Appendix A). The correctness of cluster identification and labelling was verified by comparing the components of DLPs between clusters with a one-way analysis of variance.

### 2.6. Nutrition Knowledge

Nutrition knowledge score was determined on the basis of eighteen questions. The respondents were asked about nutrition based on questions developed by Whati et al. [28] and adapted to Polish conditions and education. The correct answer was scored with 1 point, wrong or “I don’t know” answers or missing data were scored with 0 points. Points were summed up for each respondent (range 0 to 18 points). Based on tertile distribution, the respondents were divided into three nutrition knowledge categories: bottom (0–4 points), middle (5–7 points) and upper tertile (8–18 points).

### 2.7. Socioeconomic Status

Socioeconomic status was determined by the Family Affluence Scale (FAS). The FAS is a measure of material affluence derived from household characteristics. The scale is composed of four questions and was described by the Polish team of the Health Behaviour of School-aged Children (HBSC) study [29]. Answers were scored with 0 to 2 points. Some of the answers were scored with the same number of points, according to the points given below in brackets:Question: ‘Does your family own a car, van or truck?’ Answers: no (0 points); yes, one (1 point); yes, two or more (2 points).Question: ‘In recent years, how many times per year did you go away on holiday with your family?’ (necessary examples were given) Answers: not at all (0 points); once (1 point); twice (2 points); more than twice (2 points).Question: ‘Do you have your own bedroom for yourself?’ Answers: no (0 points); yes (1 point).Question: ‘How many computers or laptops or tablets does your family own?’ Answers: none (0 points); one (1 point); two (2 points); more than two (2 points).

The points were summed up for each respondent (range 0 to 7 points). Based on quartile distribution, the respondents were divided into three FAS categories: low (0–4 points; <25th quartiles), moderate (5–6 points) and high (7 points; ≥75th quartiles).

### 2.8. Adiposity Measures

The measurements of body weight (kg), height (cm) and waist circumference (WC, cm) were taken according to International Standards for Anthropometric Assessment (2001), all recoded with a precision of 0.1 kg or 0.1 cm, respectively. Professional devices and measuring tape were used (the same type across all research centres): for height—a portable telescopic measuring rod (SECA 220, Hamburg, Germany), for weight—an electronic digital scale (SECA 799, Hamburg, Germany), for waist circumference—a stretch-resistant tape that provides a constant 100 g tension (SECA 201, Hamburg, Germany). All measurements were taken in light clothing and without shoes. Body mass index (BMI, kg/m^2^) and waist-to-height ratio (WHtR) were calculated. To identify overweight/obesity (general adiposity), two standards were applied to categorize BMI-for-age with sex-specific cut-offs according to: (1) international standards (International Obesity Task Force, IOTF), 2012 (BMI ≥ 25 kg/m^2^) [30]; (2) Polish standards, 2010 (BMI ≥ 85th percentiles) [31]. As a central obesity measure (central adiposity), the waist-to-height ratio ≥ 0.5 was used [32].

### 2.9. Ethical Approval

This study was approved by the Bioethics Committee of the Faculty of Medical Sciences, University of Warmia and Mazury in Olsztyn in 2010, Resolution No. 20/2010. Informed consent was provided by teenagers’ parents or legal guardians.

### 2.10. Statistical Analysis

Categorical variables were presented as a sample percentage (%) and continuous variables as medians with an interquartile range. The differences between groups were verified with a chi-square test (categorical variables) or a Mann-Whitney test (continuous variables). Before statistical analysis, the normality of the distribution of the variables was checked with a Kolmogorov-Smirnov test.

Logistic regression modelling was applied to assess:(i)the adherence to chosen a DLP by sociodemographic factors and nutrition knowledge in respect to a referent DLP—the following categorical variables were used as predictors (independent variables): gender (girls, reference: boys), age (12 or 13 years, reference: 11 years), residence (urban, reference: rural), Family Affluence Scale (moderate or high, reference: low), nutrition knowledge score (middle or upper tertile, reference: bottom tertile);(ii)the chance to fall in the adiposity category (central obesity or overweight)—two opposite DPLs were used as predictors while others were used as a referent (i.e., in total, three various pairs of DPLs were analysed).

The odds ratios (ORs) and 95% confidence intervals (95% CIs) were calculated. Crude and adjusted models were created. Confounders included gender, age (years), residence (rural, urban), FAS (points) and nutrition knowledge score (points), excluding the modelled variable from confounders set, respectively. The significance of ORs was assessed by Wald’s statistics. For all tests, *p* < 0.05 was considered as significant. Analyses were performed using Statistica software (version 13.1 PL; StatSoft Inc., Tulsa, OK, USA; StatSoft, Krakow, Poland).

## 3. Results

### 3.1. Dietary-Lifestyle Patterns Characteristic

Three DLPs were identified, labelled as a Prudent-Active, Fast-food-Sedentary and notPrudent-notFast-food-lowActive (Appendix A). Teenagers from the Prudent-Active pattern (29.3% of the sample) were characterised by the highest frequency consumption of vegetables, fruits, fruit or vegetable juices, dairy products and fish. Most of them consumed a breakfast or a school meal every day, had vigorous physical activity at school or leisure time and screen time <2 h/day (Table 1). This pattern was considered as health-promoting due to a diet consisting of foods recommended for consumption, regularity of meals consumption and active lifestyle. In contrast, teenagers from the Fast-food-Sedentary pattern (13.8%) were characterised by the highest frequency consumption of fast foods, sweetened beverages, energy drinks and sweets, and most of them skipped breakfast or a school meal and had screen time ≥10 h/day. This pattern was considered as adiposity-promoting due to a diet consisting of food recommended to be limited with irregular meals consumption and non-active lifestyle. The notPrudent-notFast-food-lowActive pattern was represented by most of the teenagers (56.9%) and did not have any special characteristics, although to some extent it was similar to two DPLs. The notPrudent-notFast-food-lowActive pattern was similar to the Fast-food-Sedentary pattern with a low frequency consumption of vegetables, fruits, fruit or vegetable juices and fish, frequent skipping of breakfast or a school meal as well as low physical activity at school or leisure time and was similar to the Prudent-Active pattern with a low frequency consumption of fast foods, sweetened beverages, energy drinks and sweets.

### 3.2. Association between Dietary-Lifestyle Patterns, Sociodemographic Factors and Nutrition Knowledge

The Prudent-Active pattern had the highest percentage of girls (58.1%) and 11-year-old teenagers (20.0%) and a higher nutrition knowledge score (median 7.0 points). In contrast, the Fast-food-Sedentary pattern had the highest percentage of boys (59.3%) and 13-year-old teenagers (13.1%), and the lowest nutrition knowledge score (median 5.0 points) and FAS (median 5.0 points) (Table 1). The notPrudent-notFast-food-lowActive pattern was represented by students with relatively higher nutrition knowledge scores (median 6.0 points) and FAS (median 6.0) and a similar percentage of girls (50.7%) and boys (49.3%).

The adherence to the Prudent-Active (adjusted models), when compared to the notPrudent-notFast-food-lowActive pattern, was lower by 29% (OR = 0.71, 95% CI: 0.52–0.96; *p* < 0.05) in 12-year-old and 49% (OR = 0.51, 95% CI: 0.31–0.84; *p* < 0.01) in 13-year-old than 11-year-old teenagers, and higher by 57% (OR = 1.57, 95% CI: 1.16–2.13; *p* < 0.01) in the middle and 2.4 times higher (OR = 2.40, 95% CI: 1.76–3.29; *p* < 0.0001) in the upper than the bottom tertile of the nutrition knowledge score (Table 2). The adherence to the Fast-food-Sedentary (adjusted models), when compared to the notPrudent-notFast-food-lowActive pattern, was lower by 40% (OR = 0.60, 95% CI: 0.42–0.86; *p* < 0.01) in moderate than low FAS, and lower by 41% (OR = 0.59, 95% CI: 0.42–0.82; *p* < 0.01) in the middle and 58% (OR = 0.42, 95% CI: 0.27–0.65; *p* < 0.001) in the upper than the bottom tertile of the nutrition knowledge score.

In the crude model, girls were more likely to adhere to the Prudent-Active pattern (OR = 1.35, 95% CI: 1.07–1.70; *p* < 0.05) and less likely to adhere to the Fast-food-Sedentary pattern (OR = 0.67, 95% CI: 0.49–0.90; *p* < 0.05), when compared to the notPrudent-notFast-food-lowActive pattern, but both associations disappeared after the adjustment.

The adherence to the Fast-food-Sedentary (adjusted models), when compared to the Prudent-Active pattern, was lower by 38% (OR = 0.62, 95% CI: 0.43–0.89; *p* < 0.01) in girls than boys, lower by 47% (OR = 0.53, 95% CI: 0.34–0.81; *p* < 0.01) in moderate and 51% (OR = 0.49, 95% CI: 0.29–0.81; *p* < 0.01) in high than low FAS, lower by 62% (OR = 0.38, 95% CI: 0.25–0.57; *p* < 0.0001) in the middle and 84% (OR = 0.16, 95% CI: 0.10–0.26; *p* < 0.0001) in the upper than the bottom tertile of the nutrition knowledge score, and 3.2 times higher (OR = 3.20, 95% CI: 1.56–6.54; *p* < 0.01) in 13-year-old than 11-year-old teenagers.

### 3.3. Association between Dietary-Lifestyle Patterns and Adiposity Measures

The Prudent-Active pattern had the lowest percentage of students with central obesity (7.8%) or overweight/obesity (17.5% or 19.3% depending on which standard was used) (Table 1). In contrast, the Fast-food-Sedentary pattern had the highest percentage of teenagers with central obesity (15.2%) and a relatively high percentage of overweight/obese students (22.3% or 24.4% depending on standard). The notPrudent-notFast-food-lowActive pattern had a relatively high percentage of students with central obesity (13.6%) and the highest percentage of overweight/obese students (24.1% or 27.4% depending on standard).

In the Prudent-Active adjusted models, when compared to the notPrudent-notFast-food-lowActive pattern, the chance of central obesity was lower by 47% (OR = 0.53, 95% CI: 0.35–0.80; *p* < 0.01) and overweight/obesity was lower by 33% (OR = 0.67, 95% CI: 0.50–0.91; *p* < 0.05) or 38% (OR = 0.62, 95% CI: 0.47–0.84; *p* < 0.01), depending on which standard was used (Table 3). No significant associations were found between adiposity measures and Fast-food-Sedentary, when compared to the notPrudent-notFast-food-lowActive pattern, in either the crude or adjusted models. In the Fast-food-Sedentary adjusted model, when compared to the Prudent-Active pattern, the chance of central obesity was 2.22 times higher (OR = 2.22, 95% CI: 1.24–3.97; *p* < 0.01).

## 4. Discussion

Among three dietary-lifestyle patterns derived with cluster analysis, two opposite patterns were revealed, Prudent-Active and Fast-food-Sedentary, representing approx. 29% and 14% of the sample, respectively. A lower prevalence of central obesity and overweight/obesity in the Prudent-Active than the notPrudent-notFast-food-lowActive pattern (covering approx. 57% of the sample) was found, as well as a higher prevalence of central obesity in the Fast-food-Sedentary than the Prudent-Active pattern. A higher chance of adherence to the Prudent-Active pattern was shown in teenagers with higher nutrition knowledge. In contrast, a lower chance of adherence to the Fast-food-Sedentary pattern was shown in teenagers with higher nutrition knowledge or higher family affluence. More girls adhered to the Prudent-Active pattern and more boys or older teenagers adhered to the Fast-food-Sedentary pattern. No associations between dietary-lifestyle patterns and residence were found.

We identified the health-promoting dietary-lifestyle pattern (the Prudent-Active) as being clearly associated with a lower risk of obesity. This pattern was multi-component-based, i.e., composed of many dietary and lifestyle factors, as follows: high frequency consumption of health-promoting foods (vegetables, fruits, fruit or vegetable juices, dairy products and fish), regular consumption of breakfast and a meal at school and vigorous physical activity at school and during leisure time. Due to the unique composition of the Prudent-Active pattern, it is not easy to compare our findings with others. For example, in school-age Spanish children a ‘high physical activity, low sedentary behaviour, longer sleep duration, healthier diet’ lifestyle pattern was identified [14]. In 13–15-year-old Finnish adolescents, a ‘healthy lifestyle’ pattern was identified which was related to longer sleep duration, sleep quality, higher physical activity and higher fruit and vegetable consumption [15], while in fourth grade American children an ‘active, healthy eating’ pattern was identified which was characterised by a higher chance of daily fruit and vegetable consumption, participation in team sports and physical activities, weight-loss exercises and a lower consumption of soda and both high-sugar and salty/high-fat snacks [33]. In regard to obesity risk, dietary and/or lifestyle patterns, the results reported previously are not completely compatible [10,19,20,25,34,35,36,37,38]. A meta-analysis by Mu et al. [35] provides evidence that higher adherence to the ‘prudent’ or ‘healthy’ dietary patterns decreased risk of overweight/obesity while a systematic review by Janssen et al. [36] showed that overweight was not associated with fruit and vegetable consumption. In a national sample of 9–13-year-old Greeks, the pattern combining more active lifestyle with more frequent meal consumption was negatively associated with body mass index [39]. Since our study showed a clear association between the Prudent-Active pattern and obesity, it strengthens the evidence that health-promoting dietary habits and more active lifestyle are associated with lower obesity risk. Moreover, the study indicates that preventing excessive adiposity depends on many protective factors existing together as a multi-component dietary-lifestyle pattern. Thus, a holistic approach taking together dietary and lifestyle characteristics is clearly justified. Our findings provide clear evidence that future research should take into account the clustering of dietary and lifestyle behaviours to study the impact of dietary-related and lifestyle-related factors on adiposity risk in teenagers.

The study revealed no difference in adiposity between the Fast-food-Sedentary and notPrudent-notFast-food-lowActive pattern (used as referent). This may result from some similarity of these patterns. The Fast-food-Sedentary pattern was characterised by the highest frequency consumption of high-energy dense foods (fast foods) and skipping breakfast or a school meal, and the highest screen time, while the notPrudent-notFast-food-lowActive pattern was characterised by the lowest frequency consumption of health-promoting foods and also high-energy dense foods, frequent skipping of breakfast or a school meal and low physical activity.

Firstly, our study is compatible with the evidence that unhealthy dietary habits and/or sedentary lifestyle are associated with higher adiposity [17,37,38,40,41,42,43,44,45]. It has been reported that higher body mass index or overweight/obesity are associated with a ‘westernized’ dietary pattern or a ‘high animal protein and fat’ pattern in Mexican adolescents [42] or an ‘obesogenic’ pattern in Brazilian [43,44] and Spain school-age children [45] or insufficient vigorous physical activity in 11–15-year-old Americans [37]. However, a systematic review by Janssen et al. [36] showed that sweets consumption frequency was lower in overweight youth than normal-weight youth, while overweight was not associated with soft drinks consumption or time spent on the computer. Secondly, the current findings suggest that avoiding foods which favour a positive energy balance (the notPrudent-notFast-food-lowActive pattern) is insufficient to prevent obesity if, at the same time, physical activity and the consumption of health-promoting foods are low and breakfast and school meals are frequently skipped. To our knowledge, this is the first time two adiposity-promoting multi-component dietary-lifestyle patterns were identified (the Fast-food-Sedentary and the notPrudent-notFast-food-lowActive), with both presenting a similar negative association with adiposity risk in teenagers.

Since the present study is the first study investigating the association between lifestyle-dietary patterns and nutrition knowledge in Polish teenagers, a comparison of our results with others is difficult. This study clearly showed that teenagers’ higher nutrition knowledge was strongly associated with higher adherence (by 60–140%) to the Prudent-Active pattern and lower adherence (by 40–60%) to the Fast-food-Sedentary pattern. It documents the wide context of the impact of nutrition knowledge on both dietary and lifestyle behaviours. In some studies, a positive association between parental nutrition knowledge and dietary habits or single lifestyle behaviours of school-age children has been reported [21,46]. Only one study related to school children’s nutrition knowledge showed the same positive relation [47]. Our results are consistent with the previous studies but show a new aspect of teenagers’ nutrition knowledge. It can be assumed that a gain in teenagers’ nutrition knowledge can improve their lifestyle, and dietary-lifestyle education may be addressed directly to teenagers and implemented in school classes. We also note the unique role of schools in implementing diet-related and lifestyle-related education programs and also giving teenagers a chance to participate in school physical education lessons which are well-tailored to their expectations.

The study documents that the Fast-Food-Sedentary pattern was associated with lower family affluence than the Prudent-Active pattern. As we identify patterns combining both dietary and lifestyle components, our results cannot be directly compared to other results, although we can discuss the results separately with reference to dietary patterns or lifestyle characteristics. Our finding supports the view that lower family affluence is a barrier to health-promoting dietary or lifestyle behaviours. For example, a worse diet that was poor in fruits, vegetables and/or rich in highly-processed foods and/or lower breakfast frequency consumption were more frequently identified among adolescents with lower socioeconomic status from different countries, for example, the USA [48], Lithuania [34,49], the Czech Republic [50] and France [51]. It is well-documented across the world that low-energy dense foods (e.g., vegetables and fruits) are more expensive and less affordable for lower income families, resulting in disparities in access to a healthy diet [50,52,53]. High-energy dense foods, such sweets and fast foods, are generally cheaper than low-energy dense foods, so their consumption is higher in children from poorer families, especially with lower parental education [54,55]. In regard to lifestyle, positive aspects of higher family affluence were reported when children were given greater access to extracurricular physical activity and participation in team sports [56,57]. However, one contradictory finding was described. Canadian children 7–11-year-old from less affluent families were less likely to use a computer, but also less likely to participate in vigorous physical activities [58]. We hypothesize that the Fast-Food-Sedentary pattern of Polish teenagers is a result of time spent in front of the TV screen, as a cheap model of spending leisure time [22,59].

We found the clear association of gender with dietary-lifestyle patterns. This result is in line with previously reported results in regard to dietary habits, but inconsistent in regard to lifestyle. Most of the studies reported health-promoting dietary habits, but lower physical activity in girls than boys [2,3,4]. However, one study covering Polish adolescents from less urbanized regions showed that the ‘westernized Polish’ dietary pattern was more often represented by the boys who spent their free time being sedentary [60]. Our study demonstrated that girls had higher adherence to health-promoting dietary habits in combination with having a more active lifestyle (the Prudent-Active pattern) and, in contrast, boys had a higher adherence to the unhealthy dietary habits in combination with having a sedentary lifestyle (the Fast-Food-Sedentary pattern). To our knowledge, for the first time, vigorous physical activity in girls was demonstrated. This can be explained by the clustering of dietary and lifestyle behaviours in one analysis. It allowed new aspects of teenagers’ behaviours to be identified in respect to gender, compared to studies considering only single factors of diet or lifestyle or only focused on dietary patterns or lifestyle patterns [61,62,63]. Our results require further research to confirm these findings across a wide age range of Polish students as well as in other European teenagers.

The study confirmed the relationship between age and behaviours, showing the worse dietary habits (the Fast-food Sedentary pattern) in older teenagers (13-year-olds in comparison to 11-year- olds). Younger teenagers (around the age of 10–12) are still under strong influence from parents and partake in family meals, but this tendency declines as they get older [38]. Older teenagers have a lower preference for fruit, vegetables and other health-promoting foods [64,65], and are at higher risk of unhealthy dietary habits (e.g., often choose to eat fast foods, processed foods and sweets) and their diet migrated with age from a healthy to an industrialized food pattern [25]. This results from greater freedom in making food choices and the greater pressure from multiple sources around themselves, including peers [8,9].

An unexpected finding of the study was the lack of association of dietary-lifestyle patterns with residence. Across the world, rural residence is linked with lower parental education and lower family affluence and all these factors taken together result in lower socioeconomic status [66]. It is well-documented that rural residence was associated with worse dietary habits and lower adherence to dietary recommendations [67,68]. Moreover, more vigorous physical activity in adolescents from the countryside than urban areas has been reported [69]. There are two possible explanations. The clustering of many factors and/or reverse causation of some of them could have caused a lack of significant association between dietary-lifestyle patterns and residence. Secondly, progressive globalization could blur the differences between the city and the countryside through supermarket expansion, increased access to food and a change in the ways of spending leisure time [9]. Such changes could equalize the living conditions of Polish teenagers. This statement has been confirmed by several studies. About ten years ago Lazarou and Kalavana [70] found significant differences between Cypriot children from urban and rural areas due to the consumption of more traditional dishes and less fast-food by rural children. Recently, in Polish adolescents from less urbanized regions, it was shown that rural residence was not a barrier to a well-composed diet [60]. Similarly, no impact of place of residence on dietary patterns in children from public schools in Brazil was observed [24].

### Strengths and Limitations

The strength of the study is its relatively large sample (above 1500). Although the sample was not randomly selected, it covers the entire territory of Poland and widely reflects the demographic-social diversity of Poles, thus forming a good basis for generalizations. The ratio of rural/urban area in our sample (40.4/59.6) was similar to the Polish population (39.9/60.1) and the ratio of boys/girls in our sample (48.5/51.5) was also similar to the population of 10–14-year-old Polish adolescents (51.2/48.7) [71]. We applied simple measures of general and central adiposity which are appropriate for a comprehensive assessment of high adiposity levels and are widely used in large epidemiological studies, despite having some interpretative limitations [14,15,32,63].

The main limitation of the study is the use of a questionnaire to collect dietary and lifestyle data. The questionnaire contained a short list of food items (nine items) and simple questions related to sedentary and active behaviours. There is evidence of many advantages to the use of brief tools, for example, high reproducibility of simple questions, the possibility to rank respondents into categories of habitual food consumption and identify dietary patterns, the possibility to assess compliance with dietary or lifestyle recommendations as well as low cost and quick, easy administration [72]. Future studies should consider the use of a long form of the FFQ (containing more food items) to fully describe the teenagers’ dietary behaviours. However, to date, there has been no validated long form of FFQ developed for Polish children or adolescents. Secondly, the use of a questionnaire causes some uncertainty due to social desirability bias. As females are more likely to provide more socially acceptable answers, it cannot be ruled out that girls under study could have reported their own dietary and lifestyle behaviours as being closer to the recommendations than in reality [73,74,75].

## 5. Conclusions

Using a holistic approach, the study identified a set of characteristics decreasing the risk of general and central adiposity in teenagers, which includes health-promoting behaviours. Clear benefits in reducing adiposity risk were found in teenagers who frequently consumed health-promoting foods (vegetables, fruits, fruit or vegetable juices, dairy products and fish), regularly consumed breakfast and a meal at school and were vigorously active at school and leisure time (the Prudent-Active pattern). On the other hand, adiposity-promoting was a dietary-lifestyle pattern characterized by lower frequency consumption of both health-promoting foods and fast foods, frequent skipping of breakfast or a school meal and low physical activity. Such a pattern increased obesity risks similarly to another pattern characterized by the frequent consumption of fast foods and a sedentary lifestyle. Therefore, avoiding high-energy dense foods is insufficient to prevent obesity, if physical activity and frequency consumption of health-promoting foods are low and breakfast and a school meal are frequently skipped. 

The results highlight the importance of the nutrition knowledge of teenagers in shaping their health-promoting dietary habits and active lifestyle to decrease adiposity risk, and negative aspects of lower family affluence which promotes unhealthy behaviours related to both diet and lifestyle. The study supports evidence that health-promoting activities should be directed to Polish teenagers from less affluent families and should be particularly focused on increasing the consumption of health-promoting foods, improving regular meal consumption and spending more time actively. Further studies may explore gender-related dietary-lifestyle patterns to confirm our result that girls tended to display a complex of health-promoting behaviours related to diet and lifestyle, in contrast to boys who tended to display a complex of unhealthy behaviours by eating more fast foods and spending more sedentary time in front of a screen.

## Figures and Tables

**Figure 1 nutrients-10-01988-f001:**
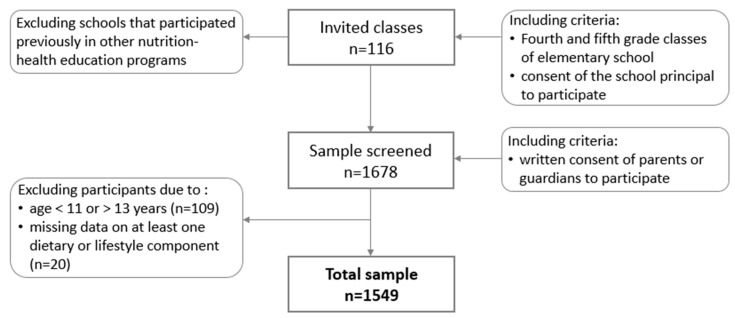
Sample collection.

**Table 1 nutrients-10-01988-t001:** Sample characteristics by dietary-lifestyle patterns (% of the sample or median (interquartile range)).

Variables	Total Sample	Prudent-Active	Fast-Food-Sedentary	notPrudent-notFast-Food-LowActive	*p*-Value
Sample size	1549	454	214	881	
Sample percentage	100.0	29.3	13.8	56.9	
Gender					***
boys	48.5	41.9	59.3	49.3	
girls	51.5	58.1	40.7	50.7	
Age (years)					*
11	16.6	20.0	13.6	15.6	
12	73.5	72.3	73.3	74.2	
13	9.9	7.7	13.1	10.2	
Residence					ns
rural	40.4	43.6	39.7	38.9	
urban	59.6	56.4	60.3	61.1	
Family Affluence Scale (points)	6.0 (5.0–7.0)	6.0 (5.0–7.0)	5.0 (4.0–6.0)	6.0 (5.0–6.0)	***
Family Affluence Scale					***
low	24.4	24.0	35.0	20.4	
moderate	49.9	52.0	43.5	48.6	
high	25.7	24.0	21.5	31.0	
Nutrition knowledge score (points)	6.0 (4.0–8.0)	7.0 (5.0–9.0)	5.0 (3.0–7.0)	6.0 (4.0–8.0)	****
Nutrition knowledge score (tertiles)					****
bottom	30.9	32.3	49.1	19.6	
middle	39.2	40.5	35.5	38.3	
upper	29.9	27.2	15.4	42.1	
Central obesity ^a^	12.1	7.8	15.2	13.6	**
BMI-for-age by international standards, 2012 ^b^					*
thinness	9.4	8.3	9.1	10.1	
normal weight	66.0	72.4	66.5	62.6	
overweight/obesity	24.6	19.3	24.4	27.4	
BMI-for-age by Polish standards, 2010 ^c^					ns
thinness	3.7	3.9	4.6	3.3	
normal weight	74.5	78.6	73.1	72.6	
overweight/obesity	21.9	17.5	22.3	24.1	
**Components of dietary-lifestyle patterns**					
Breakfast consumption (days/week)					****
<1	5.7	3.1	15.0	4.9	
1 to 3	11.8	8.6	18.7	11.8	
4 to 6	12.6	7.5	14.5	14.6	
every day	69.9	80.8	51.8	68.7	
School meal consumption (days/week)					****
<1	5.7	1.1	10.7	6.8	
1 to 2	7.3	1.1	18.7	7.7	
3 to 4	18.1	12.1	19.2	20.9	
every school day	68.9	85.7	51.4	64.6	
Frequency of consumption of (times/day)					
vegetables	0.43 (0.43–1.00)	1.00 (0.79–1.00)	0.43 (0.14–0.79)	0.43 (0.14–0.79)	****
fruits	0.79 (0.14–1.00)	1.00 (0.79–2.00)	0.43 (0.14–0.79)	0.43 (0.14–0.79)	****
fruit or vegetable juices	0.43 (0.14–1.00)	1.00 (0.43–1.00)	0.43 (0.14–0.79)	0.43 (0.14–0.79)	****
dairy products	0.79 (0.43–1.00)	1.00 (0.79–1.00)	0.79 (0.43–1.00)	0.43 (0.43–0.79)	****
fish	0.06 (0.06–0.14)	0.14 (0.06–0.43)	0.06 (0.00–0.14)	0.06 (0.06–0.14)	****
fast foods	0.06 (0.06–0.14)	0.06 (0.00–0.06)	0.43 (0.06–0.43)	0.06 (0.06–0.06)	****
sweetened beverages	0.43 (0.06–0.43)	0.14 (0.06–0.43)	1.00 (0.43–2.00)	0.14 (0.06–0.43)	****
energy drinks	0.00 (0.00–0.00)	0.00 (0.00–0.00)	0.06 (0.00–0.43)	0.00 (0.00–0.00)	****
sweets	0.43 (0.14–0.79)	0.43 (0.14–0.79)	1.00 (0.43–1.00)	0.43 (0.14–0.43)	****
Screen time (h/day)					****
<2	46.2	59.3	15.4	46.9	
2 to <4	34.5	31.9	26.7	37.7	
4 to <6	11.5	5.9	26.6	10.7	
6 to <8	4.3	2.0	11.7	3.6	
8 to <10	1.4	0.7	7.0	0.5	
≥10	2.1	0.2	12.6	0.6	
Physical activity at school ^d^					****
low	5.6	1.1	10.3	6.7	
moderate	48.7	36.1	50.0	54.9	
high	45.7	62.8	39.7	38.4	
Physical activity at leisure time ^e^					****
low	9.6	2.2	21.5	10.4	
moderate	40.4	26.4	40.2	47.7	
high	50.0	71.4	38.3	41.9	

Sample size may vary in variables due to missing data. ^a^ Central obesity identified as waist-to-height ratio ≥0.5 [32]. BMI: body mass index. BMI-for-age categorized with sex-specific cut-offs according to: ^b^ international standards (International Obesity Task Force, IOTF), 2012 [30] as follows: thinness BMI < 18.5 kg/m^2^; normal weight BMI = 18.5 to 24.9 kg/m^2^; overweight/obesity BMI ≥ 25 kg/m^2^, ^c^ Polish standards, 2010 [31] as follows: thinness BMI<5th percentiles; normal weight BMI = 5 to <85th percentiles; overweight/obesity BMI ≥ 85th percentiles. ^d^ Physical activity at school: low (most of the time in a sitting position, in class or on breaks), moderate (half the time in a sitting position and half the time in motion), vigorous (most of the time on the move or in classes related to high physical exertion). ^e^ Physical activity at leisure time: low (more time spent sitting, watching TV, in front of a computer, reading, light housework, short walks totalling up to 2 h a week), moderate (walking, cycling, gymnastics, working at home or other light physical activity performed 2–3 h/week), vigorous (cycling, running, working at home or other sports activities requiring physical effort over 3 h/week). Statistically significant: * *p* < 0.05, ** *p* < 0.01, *** *p* < 0.001, **** *p* < 0.0001; ns—not significant.

**Table 2 nutrients-10-01988-t002:** Odds ratios (95% confidence interval) for dietary-lifestyle patterns by sociodemographic factors and nutrition knowledge.

Variables	Prudent-Active (Ref.: notPrudent-notFast-Food-LowActive)		Fast-Food-Sedentary (Ref.: notPrudent-notFast-Food-lowActive)		Fast-Food-Sedentary (Ref.: Prudent-Active)	
	Crude Model	Adjusted Model	Crude Model	Adjusted Model	Crude Model	Adjusted Model
Girls (ref.: boys)	1.35 *	1.24	0.67 **	0.73	0.49 ****	0.62 **
	(1.07; 1.70)	(0.98; 1.57)	(0.49; 0.90)	(0.54; 1.00)	(0.35; 0.69)	(0.43; 0.89)
Age						
12 years (ref.: 11 years)	0.76	0.71 *	1.13	1.18	1.50	1.56
	(0.56; 1.02)	(0.52; 0.96)	(0.73; 1.75)	(0.75; 1.85)	(0.95; 2.38)	(0.94; 2.59)
13 years (ref.: 11 years)	0.59 *	0.51 **	1.47	1.56	2.51 **	3.20 **
	(0.37; 0.94)	(0.31; 0.84)	(0.82; 2.64)	(0.84; 2.88)	(1.31; 4.83)	(1.56; 6.54)
Urban residence (ref.: rural)	0.82	0.80	0.97	0.99	1.17	1.27
	(0.66; 1.04)	(0.63; 1.02)	(0.70; 1.33)	(0.73; 1.35)	(0.84; 1.64)	(0.88; 1.84)
Family Affluence Scale						
moderate (ref.: low)	1.10	1.03	0.57 **	0.60 **	0.52 ***	0.53 **
	(0.82; 1.48)	(0.76; 1.40)	(0.40; 0.81)	(0.42; 0.86)	(0.35; 0.77)	(0.34; 0.81)
high (ref.: low)	1.51 *	1.32	0.61 *	0.67	0.40 ****	0.49 **
	(1.09; 2.10)	(0.94; 1.85)	(0.40; 0.92)	(0.44; 1.02)	(0.26; 0.63)	(0.29; 0.81)
Nutrition knowledge score (tertiles)						
middle (ref: bottom)	1.56 **	1.57 **	0.58 **	0.59 **	0.37 ****	0.38 ****
	(1.15; 2.10)	(1.16; 2.13)	(0.41; 0.80)	(0.42; 0.82)	(0.25; 0.55)	(0.25; 0.57)
upper (ref: bottom)	2.55 ****	2.40 ****	0.37 **	0.42 ***	0.15 ****	0.16 ****
	(1.88; 3.46)	(1.76; 3.29)	(0.18; 0.77)	(0.27; 0.65)	(0.09; 0.23)	(0.10; 0.26)

Odds ratios adjusted for confounders: gender, age (years), residence (categorical variable), Family Affluence Scale (points), nutrition knowledge score (points), excluding the modelled variable from confounders set, respectively. Statistically significant: * *p* < 0.05; ** *p* < 0.01; *** *p* < 0.001; **** *p* < 0.0001.

**Table 3 nutrients-10-01988-t003:** Odds ratios (95% confidence interval) for adiposity measures by dietary-lifestyle patterns.

Dietary-Lifestyle Patterns	Central Obesity ^a^ (Ref: Lack)	Overweight/Obesity (Ref: Normal Weight)	
		International Standards, 2012 ^b^	Polish Standards, 2010 ^c^
Prudent-Active (ref.: notPrudent-notFast-food-lowActive)			
Crude model	0.54 **	0.61 ***	0.67 **
	(0.36; 0.80)	(0.46; 0.81)	(0.50; 0.90)
Adjusted model	0.53 **	0.62 **	0.67 *
	(0.35; 0.80)	(0.47; 0.84)	(0.50; 0.91)
Fast-food-Sedentary (ref.: notPrudent-notFast-food-lowActive)			
Crude model	1.14	0.84	0.92
	(0.74; 1.76)	(0.58; 1.21)	(0.64; 1.34)
Adjusted model	1.08	0.82	0.92
	(0.69; 1.70)	(0.56; 1.19)	(0.63; 1.34)
Fast-food-Sedentary (ref.: Prudent-Active)			
Crude model	2.12 **	1.37	1.38
	(1.25; 3.58)	(0.91; 2.07)	(0.90; 2.09)
Adjusted model	2.22 **	1.34	1.34
	(1.24; 3.97)	(0.86; 2.11)	(0.84; 2.13)

Odds ratios adjusted for confounders: gender, age (years), residence (categorical variable), Family Affluence Scale (points), nutrition knowledge score (points). **^a^** Central obesity identified as waist-to-height ratio ≥0.5 [32]. Overweight/obesity and normal weight identified as BMI-for-age categorized with sex-specific cut-offs according to: **^b^** international standards (International Obesity Task Force, IOTF), 2012 [30] as follows: overweight BMI ≥ 25 kg/m^2^, normal weight BMI = 18.5 to 24.9 kg/m^2^; **^c^** Polish standards, 2010 [31] as follows: overweight/obesity BMI ≥ 85th percentiles, normal weight BMI = 5 to <85th percentiles. Statistically significant: * *p* < 0.05, ** *p* < 0.01; *** *p* < 0.001.

## Data Availability

Due to ethical restrictions and participant confidentiality, data cannot be made publicly available. However, data from the ABC of Healthy Eating study are available upon request, for researchers who meet the criteria for access to confidential data. Data requests can be sent to ABC of the Healthy Eating study coordinator (Jadwiga Hamulka).

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
