# Peer review of "Prudent-Active and Fast-Food-Sedentary Dietary-Lifestyle Patterns: The Association with Adiposity, Nutrition Knowledge and Sociodemographic Factors in Polish Teenagers—The ABC of Healthy Eating Project"

_nutrients, 2018, doi:10.3390/nu10121988_

Round 1
Reviewer 1 Report
This study evaluates the clustering of dietary and physical activity behaviors within a large sample of Polish teenagers, and associations between patterned behaviors and nutrition knowledge and measures of adiposity.
The authors should address the following to improve the manuscript.
1) Do not use BMI cut points to define overweight and obesity in adolescents; these are not valid in this population. The authors should use the WHO BMI-for-age measures: https://www.who.int/childgrowth/standards/bmi_for_age/en/. Also, please list out the equipment used to measure heights and weights: company, model number, etc.
2) Please clarify how nutrition knowledge is being treated in these analyses. Some parts of the manuscript list refer to it as a primary predictor and in some parts it is referred to as a confounder. In general, I would suggest reframing your methods section so that you consider this paper to include a two step analysis: in the first step, you are identifying the dietary/lifestyle patters and associations between sociodemographics/nutrition knowledge and in the second step you are identifying associations between dietary/lifestyle patterns and obesity, adjusting for sociodemographics and nutrition knowledge.
3) Further describe the measures used in these analyses, especially the measure for SES. At the very least the authors should state what is included in this scale? Also describe how measures were standardized (lines 183-84).
4) Make sure that the results highlighted in the discussion section are discussed in the results section as well.
5) The authors should discuss the results from this work in the context of research in other countries. For example, Cutler et al 2009 and 2011 review dietary patterns/pa patterns in adolescents in the US. There are a lot more papers here: https://scholar.google.com/scholar?q=dietary+and+physical+activity+patterns+in+adolescence&hl=en&as_sdt=0&as_vis=1&oi=scholart. How is what the authors found the same or different from these papers? What does this study add?
6) The authors should note the limitations of using self-reported measures of diet and physical activity and how some of the gender differences in reported findings may have been due to social desirability biases in females vs. males.
7) The authors also use some difficult to interpret phrasing throughout the manuscript. For example, what does "pro-healthy" mean? Health promoting? The first sentence in the introduction is also confusing, as is the statement in lines 83-85. What does "Common" pattern mean? Consider using a more descriptive name fore this pattern.
8) I would remove the power/sample size calculations from the manuscript. These aren't usually included.
9) Add demographic data from Poland as a whole to provide a comparison of how representative the adolescents in this study are, and to provide evidence for the statement in lines 355-57.
10) lines 374-76 please justify this conclusion/discuss in more detail.
11) In the strengths and limitations section, why would using more advanced measures of body composition be necessary for this type of study when BMI-for-age is sufficient/valid at the population level and other measures are very expensive. Similarly the FFQ can be used to rank individuals, which is essentially what you are doing in this cluster analysis. What does a more detailed measure of diet add here? What are you missing? Please also discuss whether the survey measures used here were previously validated.
Author Response
Dear Reviewer,
We are very excited to have been given the opportunity to resubmit the revised version of our manuscript (ID: nutrients-385135) entitled ‘Prudent-Active and Fast-food-Sedentary dietary-lifestyle patterns: an association with adiposity, nutrition knowledge and sociodemographic factors in Polish young teenagers. The ABC of Healthy Eating Project’.
We greatly appreciate the time and efforts taken by the Reviewers and the Editor to review our manuscript. We have addressed all issues indicated in the review report, and believe that the revised version can meet the journal publication requirements.
Moreover, we have included a new author (Katarzyna Wadolowska) who was responsible for statistical analysis related to implementation of BMI-for-age to calculate overweight prevalence.
Please find our responses to the Reviewer’s comments attached. The manuscript has been corrected for language errors, using professional editing (native speaker) and proof-reading service. All changes in the manuscript are highlighted in blue font.
Yours Sincerely,
Malgorzata Kostecka

Reviewer 2 Report
This study provides much needed evidence regarding dietary and lifestyle patterns among adolescents and associations with adiposity, nutrition knowledge, residence and other socio-demographic factors.
The paper is well written with a strong methodology and reporting of results.
It may be useful to include a few words in the conclusion regarding the implication of the findings for future practice and policy. For example, what could/should we be focusing on to strengthen nutrition knowledge within this age group given the impact of this on DLPs.
Author Response

(The authors gave the same response as above.)

Round 2
Reviewer 1 Report
Thank you for addressing my comments in the revised version of the manuscript. Overall, the manuscript has been strengthened as a result. Please consider making the following additional changes for clarity:
1) When describing how variables were treated in the analyses, please further clarify which analyses they were treated as predictors of interest and which analyses they were treated as confounders (i.e. lines 168-170).
2) What are the authors pointing out in lines 175-6?
3) In the discussion the authors state that it is difficult to compare their findings to other studies multiple times. It's almost always the case that it's difficult to compare findings across studies (because of differences in the populations, the tools used, etc.). I would instead focus on the similarities and differences between what you found here, and how in looking at behaviors simultaneously this study points out that the inconsistencies in findings across the existing literature may be due to the limited number of studies that consider how behaviors important for obesity and health are patterned within groups of individuals, in different aged populations in the US.
There are a couple studies that have used similar approaches:
Laska et al (2009) DOI 10.1007/s11121-009-0140-2
Huh et al (2011) doi:10.1038/oby.2010.228
Please also expand on what you mean by "due to being multi-component" lines 314-315. Not every reader will understand what you are trying to get at here.
Double check the manuscript for minor editorial/grammatical errors as well.
Author Response
Dear Reviewer,
We are very grateful that we have the opportunity to resubmit the revised version of our manuscript (ID: nutrients-385135) entitled ‘Prudent-Active and Fast-food-Sedentary dietary-lifestyle patterns: an association with adiposity, nutrition knowledge and sociodemographic factors in Polish young teenagers. The ABC of Healthy Eating Project’.
We greatly appreciate the time and efforts taken by the Reviewer and the Editor to review our manuscript. We have addressed all issues indicated in the review report, and believe that the revised version can meet the journal publication requirements.
Please find our responses to the Reviewer’s comments attached. The manuscript has been corrected for language errors, using professional editing (native speaker) and proof-reading service. All changes in the manuscript are highlighted in blue font.
Yours Sincerely,
Malgorzata Kostecka
